



# Contemporary measurements of the background open ocean tsunami spectrum using the Deep-ocean Assessment and Reporting of Tsunami (DART) stations

Sean R. Santellanes[1,*] and Diego Melgar[1]

[1]University of Oregon, Department of Earth Science, Eugene, OR, USA
[*]Correspondance to: Sean R. Santellanes, ssantel2@uoregon.edu

**Abstract.** A reference power law of $\omega^{-2}$, where $\omega$ is angular frequency, has been traditionally used to characterize the background open ocean tsunami spectrum (BOOTS) slope from a period of 10 mins to 120 mins. However, this characterization is based on data from temporary deployments of bottom pressure sensors that lasted from several weeks to 11 months and only in scattered areas of the Pacific, leaving its effects on aleatory and epistemic uncertainties of tsunami source models unconstrained. Here we measure the BOOTS slope using 1-15 years of bottom pressure recorder data sampled at 15 s from the Deep-ocean Assessment and Reporting of Tsunamis (DART) stations. We utilize probabilistic power spectral density plots to create background noise models for 34 DART stations across the Pacific basin. We find that often a simple log-linear decay does not correctly characterize the observed background spectrum. We find deviations from the expected -2 behavior with instances of it being larger or greater, with a strong seasonality signal. In addition, we plot the median power for each DART station at the periods of 120 s, 250 s, and 800 s. Lastly, a significant part of the energy in the BOOTS is from infragravity waves, we calculate their heights and their mean values for December, January, and February and June, July, and August. We found that meteorologically induced infragravity wave events are the largest factors in seasonal variations of the BOOTS slope and intercept, especially in the east Pacific. We show the typical meteorological systems that drive these events, and we connected tropical systems from off the coast of Mexico to infragravity wave events in the east and central Pacific. Finally, we found that infragravity wave events may impact small to moderate tsunamis.

## 1 Introduction

Among the challenges affecting tsunami science today is the need for accurate and expedient source models of tsunamigenic earthquakes. Studies have been been undertaken to constrain the aleatory and epistemic uncertainty of the seismic portion of tsunami source models (i.e. via GNSS (Melgar and Bock, 2015)); however, there has been less focus on constraining these uncertainties for the oceanographic portion (i.e, via ocean bottom instruments (Tsushima et al., 2009)). Rabinovich (1997) described a method to derive tsunami source spectra given by the spectral ratio of the observed coastal tsunami spectrum and





the atmospheric-induced wave spectrum convolved with the background open ocean spectrum (BOOTS) and a constant. This method is useful because, if correct, it has the ability to allow separation of path (from bathymetry/topography) and source

effects. However, its assertion that the BOOTS slope follows — consistently — a reference power law of $\omega^{-2}$, where $\omega$ is angular frequency, is itself a potential source of aleatory and epistemic uncertainty. The basis of this reference power law can be found in the interpretations of the bottom pressure recorder (BPR) data for the open ocean for deployments of a few sites that spanned 1-11 months presented by Kulikov et al. (1983) and Filloux et al. (1991).

Following those findings, the simple log-linear spectral decay shape for the BOOTS has been traditionally used for tsunami

studies focusing on the period band 10 mins to 120 mins. However, there is compelling evidence of a myriad of forcings (Webb et al., 1991; Aucan and Ardhuin, 2013; Rawat et al., 2014) that can potentially be impacting these and shorter periods of the BOOTS and potentially introduce deviations from this simple model (Kulikov et al., 1983; Filloux et al., 1991; Webb et al., 1991; Rabinovich, 1997; Rabinovich et al., 2013). The reference power law for the BOOTS has presented a precarious situation by ignoring the short periods of the tsunami spectrum. There is no change from oceanographic forcings; infragravity

waves do not appear to influence its shape; and most puzzling of all, atmospheric forcings have been assumed to have no impact despite their purported involvement in the shorter periods by earlier literature (Kulikov et al., 1983; Filloux et al., 1991). The two simplest interpretations of these observations are: (1) The BOOTS characterizes an inherent energy cascade from low frequencies to high frequencies regardless of short period forcings within it, or (2) Previous measurements lack the spatiotemporal longevity to discern the impact of these short period forcings on its shape. Although it is surprising that

no deviations have been demonstrated from a reference power law of $\omega^{-2}$. It seems certain, from the work done in this study, that variations in the BOOTS are relatively small at these spatiotemporal scales. Here we will find that, if variations in the BOOTS can be observed, we can only infer these characteristics by using instruments that are not confined to short, modest deployments with modest spatial coverage. Fortunately, instruments, such as the Deep-ocean Assessment and Reporting of Tsunamis (DART) stations, have been designed to handle the rough nature of observation in the open ocean to provide

information for tsunami early warning systems and record continuously. We leverage these instruments to measure the long-term behavior of the BOOTS on time scales of 1-10+ years, (Figure 1, Figure 2).

The tsunami community continues to make significant progress towards constraining epistemic uncertainty in source models (Mori et al., 2022). However, a potential complication these efforts may face is the lack of constraint of the open ocean background noise necessary to produce tsunami source models. Constraining this noise is necessary for understanding proba-

bilistic effects of tsunamis as they propagate from the open ocean to coastal locations. As pointed out by Rabinovich (1997), the BOOTS slope and intercept are vital for reconstructing tsunami source spectra, which allows for higher fidelity tsunami source models from the DART stations. They can then be used to diagnose — quickly – tsunami hazards and effects. It is imperative that the BOOTS slope be measured to ensure that its effects on coastal locations can be lead to reductions in epistemic uncertainty.

Here, we report that the BOOTS slope varies substantially from the $\omega^{-2}$ reference power law used by previous literature. In fact a simple log-linear power law decays is too simplified a view of the long term behavior of the BOOTS. We find that, while the power law can be suitable for some areas of the Pacific basin and for shorter time spans, it performs poorly in other





**Figure 1.** Station up-time graphs that show the time spans of data used for each DART in this study.





regions (e.g., the Gulf of Alaska and the Cascadia Subduction Zone). We find that the areas that deviate from the reference

power law are in areas where infragravity wave (IGW) production is significant during the time periods of December, January,

and February (DJF) or June, July, and August (JJA). We further show that these time periods of IGW production correspond

most likely to meteorologicaly significant events (e.g., bombogenesis, extra-tropical cyclones, etc.). Additionally, we observe

potential IGW effects on observed tsunami signal power from periods of 120 s - 800 s. We also identify several artifacts in the

PPSD, most notably a bimodal behavior, or a bifurcation, in power for periods <80 s, in some DART stations. We conclude

that this is most likely due to changes in the DART stations themselves (i.e., upgrades of the bottom pressure recorder to DART

II or DART 4G). Finally, we discuss the implications of spatiotemporal varying BOOTS slope.

## 2 Data and Methods

### 2.1 The DART station system and PPSD measurements of the BOOTS

To test the hypothesis of the $\omega^{-2}$ reference power advocated by previous literature, we utilize the DART bottom pressure

recorder (BPR) quality-controlled data provided by the National Centers for Environmental Information (NCEI) (htt, 2005). In

real-time operations, DART stations operate using event detection to trigger changes in sampling rate. This variability in sample

rates is unsuitable for this study, however, the bottom pressure recorder (BPR) component of the DART system continuosly logs

15 s sampled data. This data is periodically retrieved by NOAA oceanographic vessels and archived. We thus limit our analysis

to include the time periods where BPR data with a 15 s sampling rate are available (Figure 1). We do not use days where data

are at sampling rates > 15 s. We then ingest day long segments of the time series into the PPSD calculation following the

75 approach detailed by McNamara and Buland (2004). This approach is desirable because it does not require any cleaning or

removal of artifacts in the data. No filter or removal of tidal signals is applied to the data — data are quality-controlled. We

measure the PPSD from the periods of 30 s to 2 hrs, and then calculate the PPSDs for the 34 DART stations shown in Figure

2. We elect to use period in seconds in order to be parsimonious with the various methods applied by referenced literature and

this study.

### 2.2 Measuring temporal variations in the BOOTS

We calculate the spectral slope and intercept of the BOOTS in two week intervals with the multitaper spectral power spectral

density (MT-PSD) methods detailed in Prieto (2022). We calculate the BOOTS intercept, as it provides a simple single param-

eter reference for the amount and type of noise present in the sampled time period (Rabinovich, 1997). We choose two week

intervals to capture the apparent seasonal variations of the BOOTS and to minimize noise from daily and weekly oceanographic

85 and meteorologic phenomena. We perform a least squares fit of the BOOTS spectral slope between the periods of 12 min - 100

min. We find that, generally speaking, the PPSDs for the DART stations fit on a spectrum between two archetypal behaviors:

BOOTS with mostly linear behavior (Figure 3a) with slope values near what the earlier literature (Kulikov et al., 1983; Filloux

et al., 1991; Rabinovich, 1997) described (a shape we refer to as "convex") and those that experience a large amount of varia-



**Figure 2.** Crosses mark the locations of the DART stations used in this study. DARTs 21420 and 46407 are highlighted because they are emblematic (21320) and non-emblematic (46407) to the reference power law of $\omega^{-2}$. BOOTS slope values.



tion and departure from the log-linear model (what we refer to as "dromedary") (Figure 3b), particularly between the periods
of 120 s - 800 s. The lower bound of 12 min was selected as it is the period when the dromedary shape of the PPSD (see Figure
3b) begins to taper off, and it is outside of the upper limit of 600 s (10 min) in the IGW band described by Aucan and Ardhuin
(2013). We select an upper limit of 100 min to minimize any long period noise effects. As with the PPSD method, we use
only data when sampling rates are at 15 s. In order to minimize tsunami effects, we do not calculate the BOOTS slope when a
tsunami occurs within a two week measuring period and for the next two week period afterwards. Tsunami coda effects can be
discernible for days after tsunamigenesis due to bathymetric and coastline effects both regionally (Melgar and Ruiz-Angulo,
2018) and globally(Kohler et al., 2020). We use the tsunami database provided by the tsu to filter out tsunamis generated by
earthquakes $\geq M_w 7.0$, landslides, and volcanoes. This procedure ensures that we are measuring background noise.

## 2.3 Measurement of IGWs with DART stations

Rawat et al. (2014) utilize numerical models and in situ measurements of DART data to track IGW events across the Pacific
basin. There they show via tracking of IGW burst events that IGW production is highest in the area of the CSZ, producing
some of the largest IGW values in the Pacific basin. We use the data preparation method from Aucan and Ardhuin (2013)
which assumes that bottom pressure recordings at IGW periods are free of pressure effects from wind-driven waves and thus
that the DART BPR data can be used to extract IGW heights at those periods. Following their approach, we consider that we
are examining free monochromatic waves of wavenumber $k$ and that we can relate the bottom pressure amplitude $p_b$ to the
surface amplitude elevation $a$ via a transfer function $M$, which is a function of water depth $D$, with the equation:

$$p_b = aM = a\frac{\rho g}{cosh(kD)}, \tag{1}$$

where $\rho$ is water density and $g$ is gravity acceleration. We relate wavenumber $k$ to wave frequency $f$ via Laplace's dispersion
relation, $(2\pi f)^2 = gktanh(kD)$. We use Prieto (2022)'s multitaper code to produce power spectral densities $F_p(f)$ of the
DART BPR data. The BPR data are subject to the same constraints as to when the BOOTS slope and intercept were calculated
in section 2.2. The transfer function $M$ is applied to the power spectral densities to obtain the surface elevation spectral density:

$$E(f) = M^2 F_p(f). \tag{2}$$

We then solve for the significant IGW height, which is defined as the partially integrated spectrum:

$$H_{IG} = 4\sqrt{\int_{f_{min}}^{f_{max}} E(f)df}. \tag{3}$$

We choose the same $f_{min}$ and $f_{max}$ as Aucan and Ardhuin (2013), setting them to $8.3 \times 10^{-4}$ Hz and $1.1 \times 10^{-2}$ Hz,
respectively. As we did not remove tsunami signals prior to calculating the significant IGW heights, we remove them and the
6 following days from the dataset. For each station, we calculate the time averaged $H_{IG}$ for the time periods of JJA and DJF.





# 3 Results

## 3.1 PPSD behavior of the BOOTS

The BOOTS intersects with what is traditionally considered the IGW band in the periods from 60 s to 600 s(Webb et al., 1991; Aucan and Ardhuin, 2013; Rawat et al., 2014). Figure 3a shows the PPSD for DART 21420 — located 480 km southeast of Miyazaki-shi, Japan, Figure 2. This station's behavior is best emblematic of the behavior expected from the $\omega^{-2}$ reference power law. Its mean slope value is $2.03 \pm 0.19$, and its mean intercept value is $10^{-4.48 \pm 0.35}$ (Table S1). We observe from Figure 3a that even for this relatively "well behaved" site the PPSD is not an exact straight line; it can have non-negligible variation of as much as 5-10 dB in the 120 s - 250 s band, giving the PPSD a convex appearance.

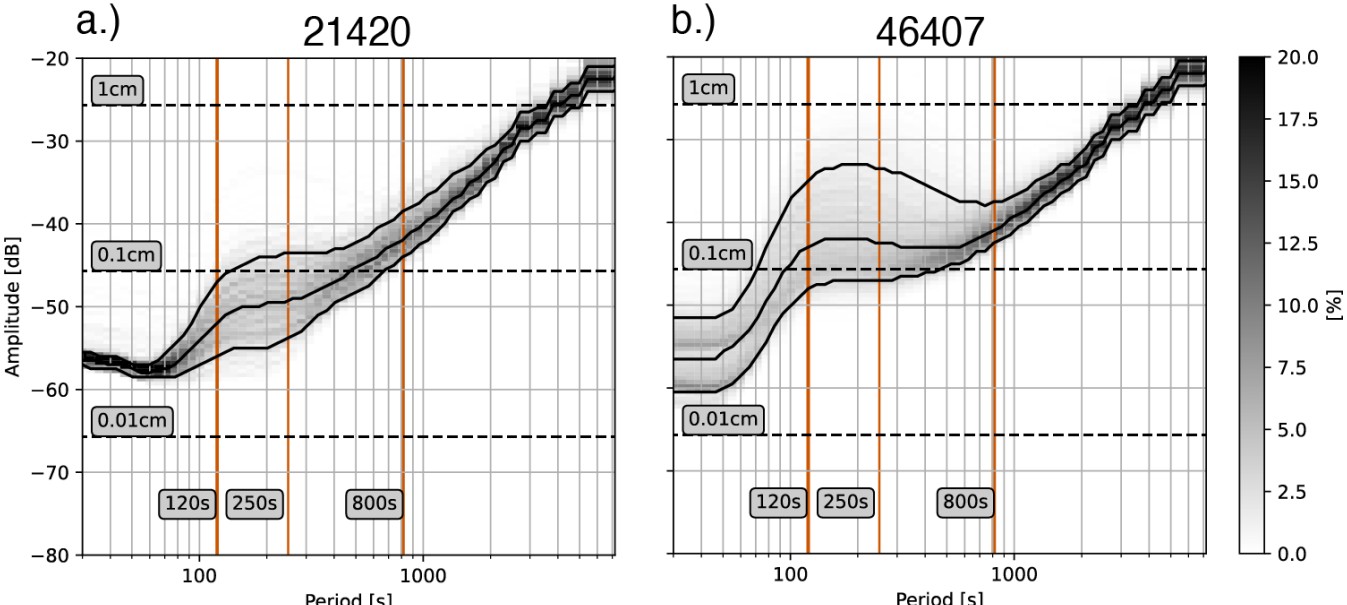

**Figure 3.** a.) Probabilistic Power Spectral Density (PPSD) plot for DART 21420 — emblematic of behavior that follows the reference power law of $\omega^{-2}$. Amplitudes corresponding to noise levels of 1 cm, 0.1 cm, and 0.01 cm are shown as dashed, black lines. Periods corresponding to 120 s, 240 s, and 800 s are delineated by solid, orange lines. b.) PPSD for DART 46407 — emblematic of behavior that does not follow the reference power law of $\omega^{-2}$. Amplitudes corresponding to noise levels of 1 cm, 0.1 cm, and 0.01 cm are shown as dashed, black lines. Periods corresponding to 120 s, 250 s, and 800 s are delineated by solid, orange lines.

Figure 3b shows the PPSD for DART 46407 — located 390 km to the west of Coos Bay, OR (Figure 2). Its behavior is dromedary in appearance — with variability as high as ~15 dB variability and hummocky between the periods 120 s - 800 s — and a significant departure from the $\omega^{-2}$ reference power law, in stark contrast of DART 21420 ( Figure 3a). Its mean slope value is $1.90 \pm 0.22$, and its mean intercept value is $10^{-4.25 \pm 0.40}$ (Table S1). Critically, Figure 3b indicates that amplitude spread for periods between 120 s - 800 s is considerable between the $10^{th}$ percentile and the $90^{th}$ percentile, which may be





evidence of external forcing. Lastly, DART 46407 has an enigmatic bifurcation in its PPSD for periods < 80 s. Something that is not observed in the PPSD of DART 21420 (3).

DART 46407's PPSD suggests that the $\omega^{-2}$ power law is a poor fit for modeling the BOOTS between the periods of 120

s - 800 s, coincident with the IGW band's 60 s - 600 s ((Aucan and Ardhuin, 2013)). This deviation from the expected trend has been noted in previous literature, resulting in them only using the reference power law from periods of only 10 mins - 120 mins (Rabinovich et al., 2011; Rabinovich and Eblé, 2015). However, it can be seen in Figure 3 that there are possible effects even in the shorter periods of that range, potentially from IGW power leaking into this band. The BOOTS extends from 2 mins - 120 mins — which includes potential IGW interactions — so it is imperative to constrain what these effects are for shorter

periods within the BOOTS.

## 3.2   Spatiotemporal variations in the BOOTS

### 3.2.1   Temporal variations in the BOOTS

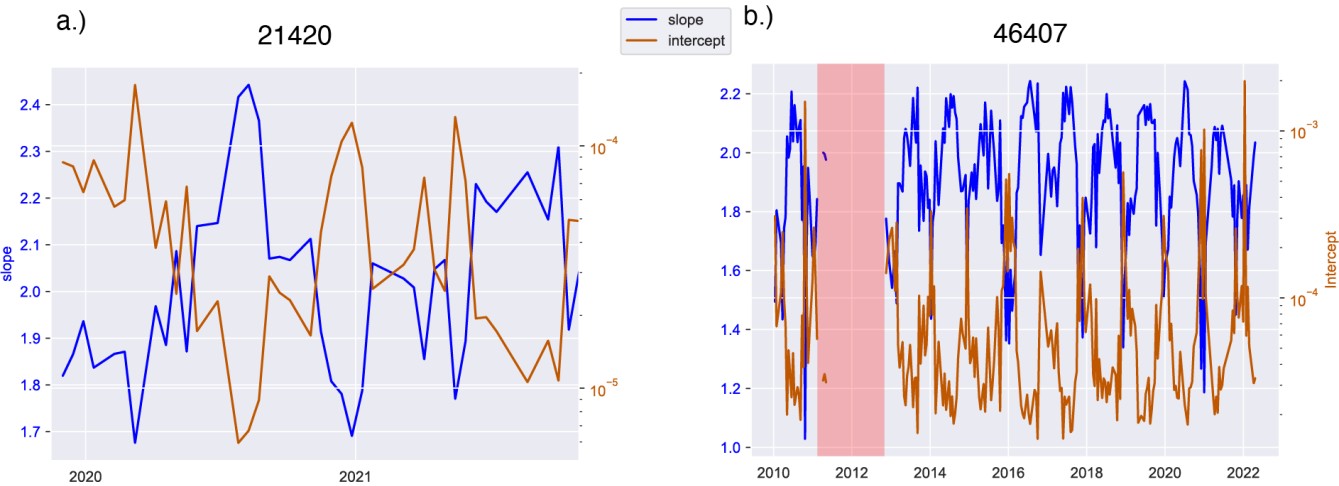

**Figure 4.** a.) 21420 time-series of BOOTS slope (blue) and intercept (orange). b.) 46407 time-series of BOOTS slope (blue) and intercept (orange). Period of low-quality data, resulting in no values for either parameter, is shown in red.

We observe from Figure 4a that DART 21420 experiences some amount of seasonal variation over the course of its deployment from December 2019 to October 2021. It achieves maximum slope values > 2.3 during the time period of JJA;

meanwhile, it obtains minimum slope values < 1.8 during the time period of DJF. We note that there is a seemingly inverse relationship between the maximum/minimum values for the BOOTS slope and intercept. Lower magnitude intercept values have been attributed to periods of less atmospheric noise (Rabinovich, 1997), meaning periods of calm weather correspond to high BOOTS slope values. We note that the period of minimum BOOTS slope values corresponds to extratropical cyclone





season for the north Pacific. However, the length of record for DART 21420 is a modest 2 years long compared to other stations
we analyze (Figure 1).

Figure 4b shows a much longer 12 year time series showcasing rich seasonal variation of bottom pressure for DART 46407.
It obtains maximum slope values in JJA > 2.2 and minimum slope values in DJF <1.4. While partially evident in Figure 4a,
Figure 4b demonstrates the seemingly inverse nature of the BOOTS slope and intercept over a much longer period of record
— January 2010 - April 2022. As discussed earlier, lower magnitude intercept values correspond with less atmospheric noise
and vice versa. Figure 4b indicates that JJA is period of seasonally calm atmospheric noise while DJF is a period of disturbed
atmospheric noise. Similar to DART 21420, DART 46407's seasonality appears to coincide with the extratropical cyclone
season in the north Pacific, which multiple sources attribute to producing IGW events (Webb et al., 1991; Aucan and Ardhuin,
2013; Rawat et al., 2014; Rabinovich and Eblé, 2015).

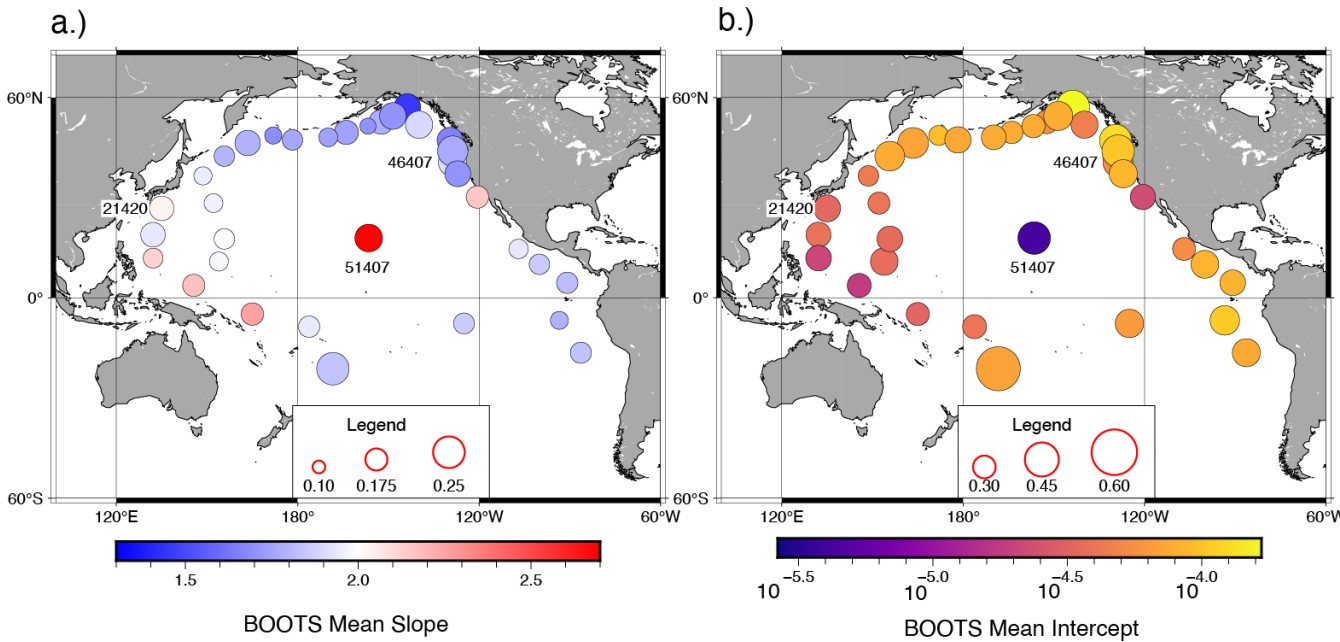

**Figure 5.** a.) Mean slope of the BOOTS across the Pacific basin for each station's entire temporal coverage. Color of the circle corresponds
to the mean value of the BOOTS slope. Circle size corresponds to the standard deviation of the slop. b.) Intercept of the BOOTS across
the Pacific basin. Color of the circle corresponds to the mean value of the intercept. Circle size corresponds to the standard deviation of the
intercept.

We demonstrate in Figure 5 that the other DART stations fall somewhere between the previously demonstrated behaviors.
We note that sites with behavior similar those similar to DART 21420 are its neighbors in the southwest Pacific. Similarly,
DART 46407's neighbors in the northeast Pacific exhibit similar behaviors. The exception to this trend is DART 51407 — the
DART nearest Hawai'i, Figure 2. This station is notable for the sheltering effect it experiences from the Big Island (Webb et al.,
1991; Rawat et al., 2014). It is not affected like other DART stations by meteorological events in the north Pacific, naturally



filtering out the noise in the BOOTS. However, we find it is susceptible to meteorologically induced IGW events from the south
Pacific and the east Pacific — west of Mexico and Central America.

### 3.2.2 Spatial variations of power in the BOOTS

As previously mentioned, DART stations 21420 and 46407 experience varying degrees of seasonal variations with other stations
falling on a spectrum between the two. We show that the stations in the north Pacific experience to varying degrees maximum
slope values during JJA and minimum slope values during DJF (Figure 6). The trend is reversed for stations from off the coast
of Mexico to off the coast of Peru and in the south-central Pacific. For these reasons, for examining the spatial variations of
the BOOTS, we split the time periods into DJF and JJA. We further focus our attention to the periods of 120 s (the start of the
tsunami band), 250 s (the middle of the dromedary hump), and 800 s (where the dromedary hump abates and end of the IGW
band).

We see from Figure 6a that the median power at the period of 120 s is in the -55 - -50 dB range for most of the north
Pacific. The southwest Pacific is where background noise is at a global minimum for the JJA and DJF months. The region of
the Cascadia Subduction Zone (CSZ) has higher power than the rest of the north Pacific, being between -50 dB - -45 dB during
JJA. The area has the highest powers for the Pacific basin being <-40 dB during DJF. The DART stations in the southern
hemisphere and off the coast of Central and South America achieve their highest power values at 120 s during JJA and lessens
only slightly during DJF. The DJF period demonstrates that there is an apparent eastward increase in power for the 120 s period,
which is not present during JJA.

For the median power at the period of 250 s, we observe a similar behavior to that of 120 s period. At this period the Aleutian
Islands and Gulf of Alaska DART stations join the stations off of the CSZ in having values <-41 dB. Whereas at 120 s, there
was a gradual increase in power, they are in the same range of -41–38 dB, Table S1. The CSZ DART stations still produce
the maximum power value of -35.7 dB at DART 46404. The DART stations east of the Izu-Ogasawara and Marianas Trenches
have higher power than those to the west. This could be due to the sheltering effect the trenches and their islands produce from
storms in the open ocean during DJF. This effect appears to be minimal.

We find that when the PPSDs are mostly no longer affected by the dromedary hump at the longer periods. For example, at
800 s a more unique spatial variation is present. In the vicinity of DART 21420, the stations closest to Japan have a higher
value than those to the south during DJF. In fact, the power is higher at the stations from 21420 all the way to the stations in the
far east Aleutian islands. We espy that the dromedary hump has not completely relaxed at the period of 800 s for these stations.
Whereas there was a minimal sheltering effect produced by the Izu-Ogasawara and Marianas Trenches during DJF in the other
periods, this effect is not present for the period of 800 s. We discern that DART 52406, in the south Pacific, is a station with
power >-39 dB in both JJA and DJF for the region. We recognize that the station's BPR is located in comparatively shallow
water (1800 m) which may play a role in why it has higher power than its neighboring stations. Generally, it follows from
Laplace's dispersion relation that the deeper a station is, the more the more the water column attenuates short period signals.
Since the depth of this BPR is shallower than those of other stations, it is more susceptible to short period noise that would
otherwise be attenuated. Hemispherical differences in power are the most apparent of all periods at a period of 800 s in JJA. As







**Figure 6.** a.) Median power in dB at a period of 120 s. Top panel is median power during JJA, and bottom panel is median power during DJF.

b.) Median power in dB at a period of 250 s. c.) Median power in dB at a period of 800 s.



with the previous periods, the CSZ DART stations continue to be the region of highest power during DJF. Finally, we pinpoint, again, that DART 51407 has low power during both JJA and DJF due to the sheltering effect of the Hawaiian Islands.

## 4 Infragravity waves in the BOOTS

Next, we show that the chief driver of the spatiotemporal variability of the BOOTS slope and intercept are most likely IGW.

### 4.1 Spatiotemporal Effects of IGWs

**Figure 7.** Infragravity wave heights for the time periods of JJA (red circles) and DJF (green circles). DART locations are shown as black dots.

We find that the highest values of $H_{IG}$ and the strongest seasonality are in the northeast Pacific within the Gulf of Alaska and in the area of the CSZ (Figure 7). $H_{IG,DJF}$ is a factor of 2-3 larger in these regions than $H_{IG,JJA}$. We observe that $H_{IG}$





values decrease westward from the Gulf of Alaska and the CSZ. Aucan and Ardhuin (2013) attribute this effect to the convex shape of the Alaska peninsula and the Aleutian Islands, where IGW free energy can disperse over a wider ocean region. Our areas of high $H_{IG}$ are in line with the values measured by Aucan and Ardhuin (2013) and Rawat et al. (2014).

    We find that, in the lower latitudes of the Pacific, the values of $H_{IG}$ are lower and have a less pronounced seasonality. From Mexico to Peru, $H_{IG}$ have a reversed seasonality signal. This result is in line with the results of Filloux et al. (1991); Aucan and

Ardhuin (2013); Rawat et al. (2014). However, we find, in addition to austral winter storms in the south Pacific, that tropical systems are another likely source of IGW events.

## 5   Data artifacts in the BOOTS

### 5.1   Tsunami signals

We have shown that there are other signals present in the BOOTS that are not tsunami signals that produce significant variations

in amplitude between the periods of 120 s - 800 s. However, what remains to be clear are potential impacts of these signals on tsunami signals. In order to do so, we pick out 3 tsunamigenic earthquakes to observe the type of signals they produce in the DART stations' PPSDs. We focus our attention on DART 46407, since it is among the stations that experiences the highest $H_{IG}$ values in the period of DJF.

    We choose the 2020 $M_w 7.8$ Simeonof, 2020 $M_w 7.6$ Sand Point, and 2021 $M_w 8.2$ Chignik earthquakes (8). The $M_w 7.8$

Simeonof and $M_w 8.2$ Chignik earthquakes both occur in July, which is when $H_{IG}$ are at their lowest values for the Aleutians and the CSZ. The $M_w 7.6$ Sand Point earthquakes occurs in October, which is when $H_{IG}$ begin to become higher leading into DJF, Figure 3. The $M_w 7.6$ Sand Point signals are shown twice, as the earthquake occurred near the end of the UTC day. We see from Figure 9 that Simeonof experiences a slight reduction in power at 250 s, and it is below the $10^{th}$ percentile of noise between 120 s - 400 s. Sand Point's apparent short period signal is slightly lower in power than that of Simeonof and Chignik.

Yet, its signal between 170 s - 1100 s is larger than either event. We note that there is no reduction in power in the signal between 120 s - 400 s like with Simeonof or Chignik. Chignik's short period power is larger than both Simeonof and Sand Point, but as mentioned previously with Simeonof, its power between 120 s - 800 s experiences a similar reduction in power.

    $H_{IG}$ values 24 hours prior to the Simeonof earthquake were between 6-7 mm. The event itself did not elevate $H_{IG}$ by a noticeable amount, even in the days immediately after the event. Whereas, $H_{IG}$ values after the Sand Point and Chignik earth-

quakes have amplitudes of 15 mm and 13 mm, respectively. The difference for Sand Point and Chignik is that the background $H_{IG}$ values were 1-2 mm higher in the days before Sand Point than in the days prior to Chignik. We discuss ,later, potential impacts of the IGW band on the BOOTS.

### 5.2   Bifurcation of PPSD in short periods

The apparent birfurcation of the PPSD for 46407, as with other stations where it occurs, at first seems complex. We investigate

whether it is the result of differing deployment locations of the DART BPR. Frequently, after a DART site is maintained or





**Figure 8.** The 2020 Simeonof rupture zone from Crowell and Melgar (2020) is shown in black, the 2021 Chignik rupture zone from the USGS-National Earthquake Information Center (NEIC) finite fault model for the event is shown in dark blue, and the rupture area for the July 15, 2023, Sand Point earthquake from the USGS-NEIC finite fault model is shown in aquamarine (Survey, 2017). The surface projection of the strike-slip plane associated with the 2020 Sand Point earthquake is delineated by a dashed red line.



**Figure 9.** DART 46407 PPSD. The 2020 M7.8 Simeonof earthquake is shown by a dash-dot-dot magenta line (UTC Day July 22, 2020). The 2020 M7.6 Sand Point earthquake is shown in blue (UTC Day October 19, 2020) and by a dash-dot crimson line (UTC Day October 20, 2020). The 2021 Chignik earthquake is shown by a dashed green line (UTC Day July 21, 2021).



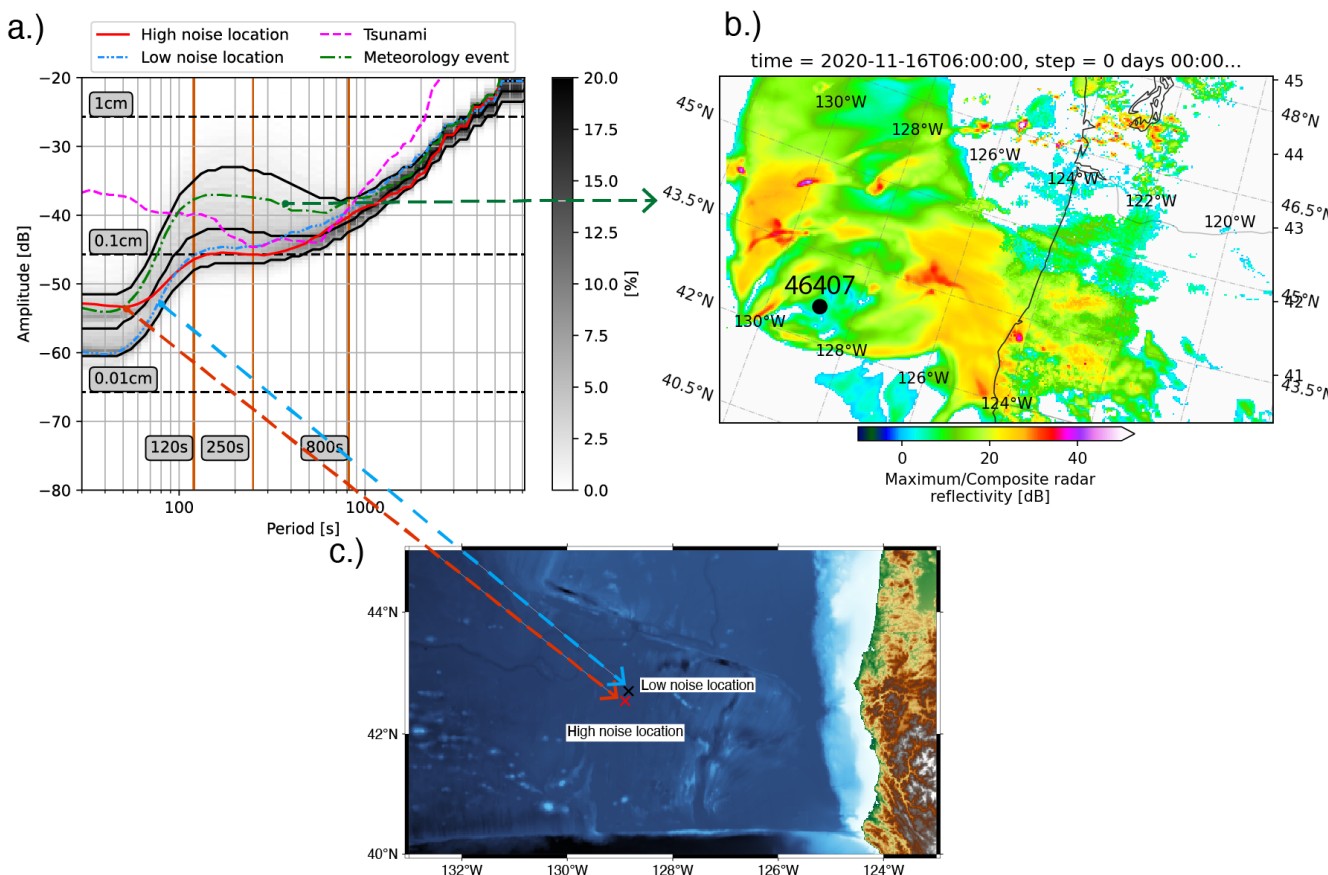

**Figure 10.** a.) PPSD for DART 46407. High noise location (red), low noise location (dash-dot-dot blue), meteorological event (dash-dot green), and tsunami (July 22, 2021 Chignik earthquake, dash magenta) PSDs are plotted for their respective occurrences. Arrows point to their respective phenomena in panels b.) and c.). b.) Meteorological event that occurred on November 16, 2020, at 0600 UTC which produced noise above the 10th percentile in the BOOTS. Maximum/Composite radar reflectivity from the 0600 UTC High Resolution Rapid Refresh (HRRR) model is shown. c.) Location of DART 46407 at a high noise location (UTC Day May 17, 2011, red x) and a low noise location (UTC day July 7, 2021, black x).





overhauled the new buoy and BPR can end up in a different location than prior to the work being done on it. We find, at maximum, that new BPR deployments are between 20-30 km of the previous deployment location. In the example of 46407, its location is on the abyssal plain — far from any complex structures, as seen in Figure 10c. We choose two days and compare their PPSD signals with each other. One when the DART was located in a high noise location (2011/05/17) and in a low noise location (2021/07/07). Ultimately, we find the solution to the bifurcation in the metadata for the BPR data. It was upgraded to a lower noise floor instrument over the span of 10 years, so the apparent bifurcation is a symptom of the sensor itself being upgraded and not due to a change in location. We find that this is the case for all BPR data that spans >5 years. The apparent bifurcation abates for periods longer than 80 s, so it does not pose an issue for examining signals that reside in the BOOTS. Although, it may pose an issue to those using the data for long term investigations of IGW events.

## 5.3 Meteorological impacts on the BOOTS

Numerous works in the literature describes that IGWs and the BOOTS are impacted by meteorological events yet refrain from showing these systems' meteorological set ups (Kulikov et al., 1983; Filloux et al., 1991; Rabinovich, 1997; Aucan and Ardhuin, 2013; Rawat et al., 2014). Here we use weather model output to connect days of high $H_{IG}$ values to specific types of meteorological systems. We choose the High Resolution Rapid Refresh (HRRR), European Centre for Medium-Range Weather Forecasts (ECMWF) weather and wave models, and the Global Ensemble Forecast System (GEFS) to investigate storm systems across the Pacific, where we have shown in earlier sections that $H_{IG}$ values are high. We utilize a variety of models due to the varying availability of model output for the time duration of our study. Model output is not always available for the time period of 2006-2022 due to storage or open data availability issues. Model output are processed using Herbie (Blaylock et al., 2017; Blaylock, 2024).

For the HRRR model, we visualize storms systems by using the maximum/composite radar reflectivity. For the ECMWF wave model, we visualize storm systems with the significant height of combined wind waves and swell forecast and geopotential height at 850 mb pressure level with 10 m winds plotted to visualize the wind field of the tropical and extratropical systems. For the GEFS, we use the mean member pressure reduced to mean sea level with 10 m winds plotted to visualize the wind field of the tropical and extratropical systems.

Figure 10b shows an archetypal, modest extratropical storm that affects the CSZ during November. These types of storm systems occur frequently during this time of the year; however, their strength is variable. This storm system was responsible for $H_{IG}$ of 11 mm for November 16, 2021. In the immediate days ahead of the storm system, $H_{IG}$ values were as high as 14 mm. These systems do not need to transit directly over the DARTs to elicit such high $H_{IG}$ values. As noted by Hanafin et al. (2012), storm track and intensity are crucial factors in IGW event generation for this region. We find that events that make landfall on the North American coast between $40°$ and $60°$ produce IGW events of varying intensity.

Figure 9a shows the ECMWF significant height of combined wind waves and swell for an extratropical storm that transits through the Gulf of Alaska during February. Storm systems that transit through this region are strong enough to elicit high $H_{IG}$ values at the CSZ DART stations. They have significant height of combined wind waves and swell of 10-12 m in the Gulf of Alaska, yet only produce heights of 3-4 m over the CSZ. As Aucan and Ardhuin (2013) and Rawat et al. (2014) note, these







**Figure 11.** a.) ECMWF wave height model at 0.5° resolution for February 10, 2022 at 12 UTC. DART 46407 is shown by a black dot. b.) ECMWF model at 0.5° resolution for February 10, 2022 at 12 UTC. Geopotential height at 850 mbar is plotted at 50 m contours. 10 m wind is shown by filled contours. c.) PPSD for DART 46407 for the meteorological event signal during August 13, 2021. The signal is shown by the dashed green line.



265 storm systems produce conditions favorable for IGWs due to their storm track and intensity. For the February 10, 2022 storm, the system's center is located at $142°W$ and $60°N$, yet it is producing widespread IGW impacts across the area, in spite of a high pressure system that has set up at $135°W$ and $45°N$, Figure 9b and 9c. Low wind speeds in the high pressure area do not efficiently generate wind waves, so the wind generated waves and swell from the extratropical system can transit across the area with little to no impact.

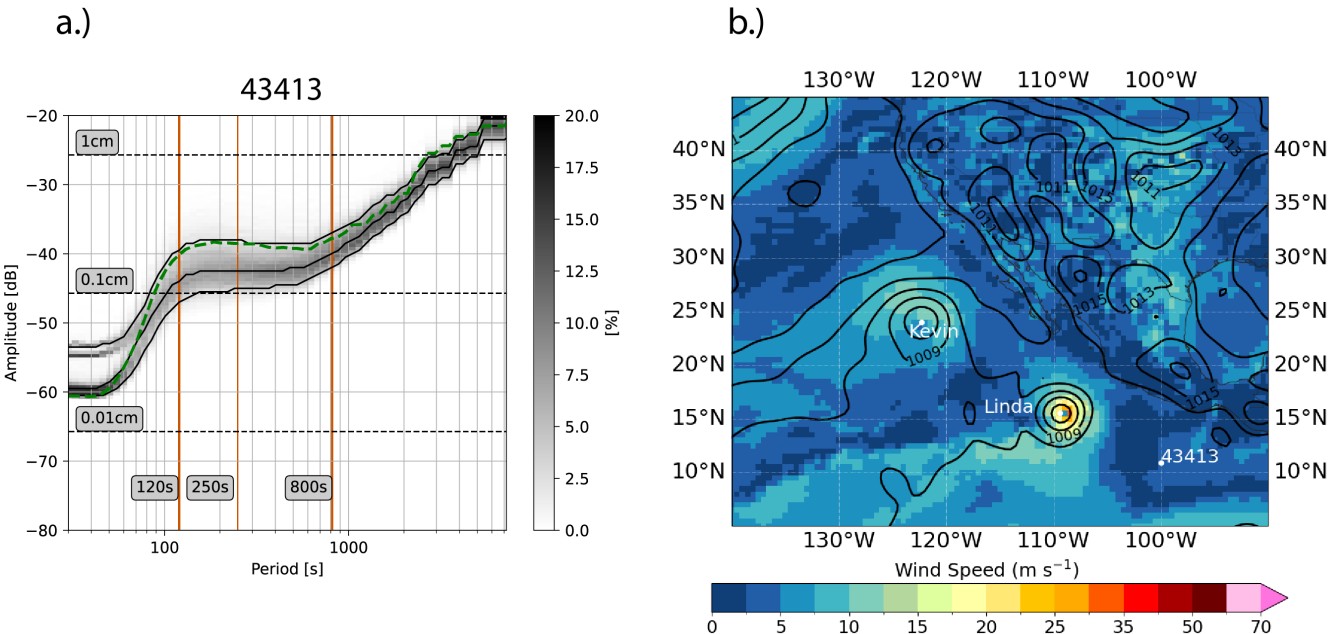

**Figure 12.** a.) PPSD for DART 43413. Meteorological event signal for August 13, 2021 is shown by the dashed green line. b.) GEFS model at $0.5°$ resolution for August 13, 2021 at 12 UTC. 10 m wind is shown to visualize the wind field of the tropical systems. Pressure reduced to mean sea level (MSL) pressure is shown by the black contours. Contours are plotted at 2 mbar intervals. DART 43413 is shown by a white dot. The approximate locations of tropical storm Kevin and Hurricane Linda's cores are shown by white dots.

270 In contrast, DARTs 43413 and 43412 show evidence for IGW events forced by tropical systems. Figure 10 shows the PPSD and PPSD signal for DART 43413 as Tropical Storm Kevin and Hurricane Linda pass to the northwest of the station. As seen with the CSZ DARTs, the stations along the Middle and South America Trenches experience higher $H_{IG}$ values when transit close to the stations. Higher $H_{IG}$ values for the DART stations off of South America are also experienced due to austral winter storms in JJA in the southern hemisphere.





## 6    Discussion

### 6.1    The magnitude of the BOOTS slope

The variety of data used in this study lead to an inescapable conclusion: the BOOTS slope does not uniformly follow a reference power law of $\omega^{-2}$, even when we limit its use to periods where IGW forcing is minimal from the PPSD. Therefore, we posit that the BOOTS varies spatiotemporally, majorly as a function of IGW forcings going from short period to long period within the BOOTS. Characteristic of this effect is the presence of a dromedary hump for periods of 120 s - 800 s at DART stations in the north and east Pacific regions. We recognize that this hypothesis is seemingly at odds with earlier literature. However, we find that certain regions of the Pacific (i.e., the southwest and south central Pacific and offshore California), where PPSD follows an convex behavior, do follow, to some degree the reference power law. In fact, variations from the reference law are smallest during March, April, May (MAM) and September, October, and November (SON) when seasonal signals in the CSZ, Gulf of Alaska, Central America, and South America are in transition seasonal maximums and minimums. Yet still, other areas, as is the case for DART 21420, experience little variations about the reference law for the whole year.

Explaining the differing behaviors between dromedary and convex background profiles has a simple solution: IGW effects. When we combine our results with the results from Aucan and Ardhuin (2013) and Rawat et al. (2014), we find that stations with dromedary PPSD shapes directly correlate with areas of intense IGW production, Figure 7. And, we find that stations with convex PPSD shapes correspond to those where sheltering from nearby bathymetric and topographic features occur (Rawat et al., 2014). Our results in Figure 7 are in harmony with earlier IGW literature.

### 6.2    Implications of meteorological impacts on the BOOTS

IGW-genesis from meteorological forcings has long been established from the work of Webb et al. (1991). Kulikov et al. (1983) even refer to the effects of transient weather systems on the noise they recorded during their expeditions. Aucan and Ardhuin (2013) and Rawat et al. (2014) add that winter storms in the Gulf of Alaska and the CSZ as being the biggest producers of high $H_{IG}$ values in the time frames considered by their studies. Hanafin et al. (2012) found that intensity, duration, and storm track are biggest indicators of large IGW bursts. We have shown that normal extratropical intensification is enough to elicit high values of $H_{IG}$ in the Gulf of Alaska and CSZ. In addition, we have shown that for stations off the coast of Mexico, Central America, and South America have high $H_{IG}$ values due to the East Pacific hurricane season in JJA. In addition elevalted values due to austral winter storms in the South Pacific.

Explanations for the high $H_{IG}$ values in the Gulf of Alaska and the CSZ have already been covered by Rawat et al. (2014). Where they found that extratropical cyclones with predominant westerly winds and waves cause significant IGW events. Figures 9 and 10 show examples of common types of extratropical storm systems. We posit that storm track and duration are the chief contributors to IGW heights in this area. These storms deepened over the Gulf of Alaska and made landfall between $45°$ and $60°$ and transited through the Gulf of Alaska over a period of 3-5 days. We found that storm systems that underwent bombogenesis — a 25 mbar drop over a period of 24 hours — prior to landfall did not produce higher than normal $H_{IG}$ values





for DJF. Often these systems deepened within 12-24 hours of landfall right off the coast such that there was not enough time to induce a wide wind field.

We posit that high values of $H_{IG}$ from Mexico to Peru follow a similar generating mechanism; however, with tropical systems rather than extratropical systems. These systems with predominant easterly winds and waves cause significant IGW bursts for the area. Figure 10 shows a common meteorological set up for the East Pacific in JJA — multiple tropical systems off the coast of Mexico. These storm systems produce winds and waves that radiate outward inducing $H_{IG} > 15$ mm for the stations in the area. These storm systems even induce seasonal high values in $H_{IG}$ for stations in the CSZ and even DART 51407 near Hawai'i, which is sheltered by the Big Island of Hawai'i from extratropical IGW events in the Gulf of Alaska and the CSZ during DJF.

Meteorological forcings that drive IGW bursts are sources of aleatory and epistemic uncertainty in the BOOTS, and, thus, tsunami source models. While it is not possible to know the exact state of these forcings when a potential tsunamigenic earthquake occurs, it is possible to forecast events that drive IGW bursts up to 7 days of lead time (Kalnay, 2002). Future work can focus on combining GFS and ECMWF wave forecasts with the PPSD models of the DARTs to futher refine the aleatory and epistemic uncertainty effects of meteorologic effects on tsunami propagation from the open ocean to coastal sites.

### 6.3 Implications of a spatiotemporally varying BOOTS

We have shown that IGWs have an impact on the BOOTS from 120 s - 800 s, and that these impacts affect the BOOTS slope even when measuring it from 12 mins - 120 mins to reduce IGW effects. We have posited, in agreement with previous literature, that extraopical and tropical systems can lead to IGW generation events. In addition, we have shown that IGW affect the BOOTS slope from 120 s - 800 s. Figure 7 shows the PPSD signatures of 3 earthquakes that produced tsunamis in the Gulf of Alaska and propagated as far as Hawai'i. The 2020 Simeonof and 2021 Chignik earthquakes occurred in July when IGW generation is at yearly low in the north Pacific. From the periods of 120 s - 800 s, these earthquakes' tsunami PPSD signatures experience a reduction in power. In contrast, the 2020 Sand Point Earthquake's tsunami PPSD signature experiences no similar reduction in power; it appears to increase through the range of periods, Figure 7. The BOOTS slope during July 2020 and July 2021 was >2.2 while the slope was <1.8 during October 2020, Figure 3.

We posit that the IGW band and the BOOTS interact with one another. This hypothesis is not extreme. As Rawat et al. (2014) notes, IGWs are only a few periods shorter than those of large tsunamis. And, both have similar propagation speeds and spatial distribution of amplitudes caused by shoaling and refraction. These similar physical characteristics may lead to interesting interactions. It is possible that the IGW band and tsunami band destructively interfere with one another when the BOOTS slope is >2 and constructively interfere when it is <2. It would therefore be imperative for tsunami source reconstruction and inversion to take into account these effects in order to obtain high fidelity tsunami source spectra and source regions.

High fidelity BOOTS noise models must take into account IGW effects for automatic detection of tsunami waves. The similar nature of IGWs to tsunamis from 120 s - 800 s, could potentially lead to false positive tsunami detections during meteorological caused IGW events or lead to false negative tsunami detections when IGW events cause large amounts of background noise.



# 7 Conclusion

We have shown that the BOOTS slope does not universally follow a reference power law of $\omega^{-2}$. We found that it experiences substantial seasonal variations in the east Pacific from the Gulf of Alaska down to Peru. The chief contributor to its variation are meteorologically induced IGW events. These events interact with the BOOTS to lower the slope value to be $< 2$. In the absence of these events, BOOTS slope values are $\geq 2$, with values being closer to 2 in areas outside of the east Pacific. We have proposed that IGW events and their absence may have some interactions with tsunami events that may affect aleatory and epistemic uncertainties for tsunami source models. We have shown that, in addition to extratropical systems in the north and south Pacific, tropical systems in the east Pacific cause IGW events that can be recorded across the Pacific. Finally, we recommend that automatic detection algorithms take into account IGW events, particularly in the Gulf of Alaska and CSZ, as they may eclipse small tsunami signals.

*Code and data availability.* DART BPR data is available at https://www.ngdc.noaa.gov/hazard/dart/. Records of tsunamis in the Pacific basin are available at https://data.noaa.gov/metaview/page?xml=NOAA/NESDIS/NGDC/MGG/Hazards/iso/xml/G02151.xml&view=getDataView. The Herbie python code is available from GitHub at https://github.com/blaylockbk/Herbie and archived on Zenodo at Blaylock (2024). The PPSDs were generated using the multitaper python package available from GitHub at https://github.com/gaprieto/multitaper and documented by Prieto (2022). Table S1 and other codes necessary for the study are available from GitHub at https://github.com/ssantellanes/BOOTS and archived on Zenodo at Santellanes (2024).

*Author contributions.* SS contributed to conceptualization, methodology, software, formal analysis, investigation, writing, reviewing, editing, and visualization. DM contributed to conceptualization, methodology, resources, reviewing, editing, supervision, and funding acquisition.

*Competing interests.* The authors declare that they have no conflict of interest.

*Acknowledgements.* We would like to thank Aaron Sweeney for helpful discussions on processing DART data. A portion of this work used code generously provided by Brian Blaylock's Herbie python package (https://doi.org/10.5281/zenodo.4567540).



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
