# Peer review of "Contemporary measurements of the background open ocean tsunami spectrum using the Deep-ocean Assessment and Reporting of Tsunami (DART) stations"

_EGUsphere, 2024_

## Referee Comment (RC1)

Review

of "Contemporary measurements of the background open ocean tsunami spectrum using the Deep-ocean Assessment and Reporting of Tsunami (DART) stations" by *Sean R. Santellanes and Diego Melgar*

This is an interesting paper. Open-ocean DART stations are working in the Pacific Ocean for more than 30 years, however, unfortunately, they are still poorly used for general oceanography and wave dynamics, except the examination of specific tsunami events. From this point of view, the submitted paper partly fills the gap. The problems discussed by the authors are far from being trivial and naturally they produce serious questions and comments that impugn the main results of the paper. Also, the authors know the literature on these problems quite well, including some old publications, they missed some important points indicated in these papers. Also, there are a few additional papers that would be useful to discuss and refer.

There are two principal comments related to the submitted ms:

(1) The main topic of the paper is the deviations of actual open-ocean bottom pressure spectra from the $\omega^{-2}$ power law. This low is determined by two factors: (a) the shape of the air pressure spectrum and (b) the cascading energy transfer from low to high frequencies. However, this law does not work and cannot work for infragravity (IG) waves that have quite different physics! The IG-waves are generated by nonlinear interaction of wind waves or swell. As was shown by *Kovalev et al.* [1991], during strong storms there is an intensive IG energy transfer from high frequencies to low frequencies, the effect is similar to "negative viscosity": the stronger the storm the more energy to the lower-frequency spectrum. Therefore, *Kulikov et al.* [1983], *Rabinovich* [1997] and *Rabinovich and Eblé* [2015] told that the $\omega^{-2}$ power law typically works only till the frequency 0.1 cycles per min (cpm), i.e. for periods > 10 min. The authors know this, they mentioned several times in their text. Nevertheless, they made all their estimates for the frequency band $8.3 \times 10^{-4}$ – $1.1 \times 10^{-2}$ Hz (for periods from 1.5 to 20 min), i.e., for the frequency band that partly is strongly affected by IG-waves and where all their results are becoming senseless!

(2) Significant part of authors' study is related to estimation of the IG energy and the "dromedar hump" (as the authors call it). However, this "hump" is a purely artificial effect of nonhydrostatic HF wave attenuation and all these results are incorrect! The hydrostatic equation is valid only for **long** (compared with the ocean depth) waves, while short (high-frequency, HF) waves strongly attenuate with depth as $1/\cosh(kh)$, where $k$ is the wave number. As a result, waves with a period of ~2 min becomes almost undetectable at depth $h = 5$ km (the approximate depth of DART 21420). *Rabinovich and Eblé* [2015] illustrate this in their Figure 13. The authors write this themselves in their Eq. (1) but then forget

about this and consider all their estimates of IG waves as real! The authors were supposed to correct their spectra using the Eq. (1) and dispersive relation in Line 107.

Minor comments:

(3) The spectral analysis is the **frequency** analysis [e.g. *Thomson an Emery*, 2014]. All dispersive relationships are normally written for **frequencies**, not for periods! The authors themselves in their Eqs. (2)–(3) indicate frequencies and estimate wave energy for the frequency band between $f_{min}$ and $f_{max}$! Despite these, all their spectra (Figures 3, 9-12) are shown as functions of **periods (?!)**. This looks very unprofessional! (Once again, look at *Thomson an Emery* [2014]). BTW, even the $\omega^{-2}$ power law is frequently called the "decay law".

(4) By the publishing rules, all variables should be written in *italic*, while all functions by regular! I.e. in Eq. (1) → cosh($kD$), in Line 107 → $gk$tanh($kD$).

(5) If the authors are talking about the $\omega^{-2}$ power law, they should show this law in their spectra.

(6) In caption to Figure 3 (Line 3) it is written 240 s, while in the figure 250 s.

(7) If the authors are talking about specific integration frequency bands of $8.3\times10^{-4} - 1.1\times10^{-2}$ Hz, they should indicate them in their spectra!

(8) The logical question: If tsunami periods are up to 2 hours (in fact, even larger!), why the authors estimate the power law for much smaller periods (up to 20 min)?!

(9) Two additional useful references when the authors are talking about DARTs are *Mofjeld* [2009] and *Mungov et al.* [2013].

(10) *Zaytsev et al.* [2016, 2017, 2021] used the $\omega^{-2}$ power law and actual background DART spectra to reconstruct the real tsunami spectra in the open ocean. The authors can select one of these papers and look at the results. The advantages and limitations of such approximation are evident.

The general conclusión: **Major revision**

References

Kovalev, P.D., Rabinovich, A.B. and Shevchenko. G.V. (1991), Investigation of long waves in the tsunami frequency band on the southwestern shelf of Kamchatka, *Natural Hazards*, 4, (2/3), 141-159.

Mofjeld, H.O. (2009), Tsunami measurements, In: *The Sea*, Vol.15, Tsunamis, (Eds. E. Bernard and A. Robinson), Harvard University Press, Cambridge, USA, pp.201-235.

Mungov, G., Eblé, M., & Bouchard, R. (2013), DART tsunameter retrospective and real-time data: A reflection on 10 years of processing in support of tsunami research and operations, *Pure and Applied Geophysics, 170*, 1369-1384. https://doi.org/10.1007/s00024-012-0477-5

Thomson, R.E., & Emery, W.J. (2014). *Data Analysis Methods in Physical Oceanography*, Third and Revised Edition. New York, Elsevier, 728 p.

Zaytsev, O., Rabinovich, A.B., & Thomson, R.E. (2016), A comparative analysis of coastal and open-ocean records of the great Chilean tsunamis of 2010, 2014 and 2015 off the coast of Mexico, *Pure and Applied Geophysics, 173*(12), 4139-4178. https://doi.org/10.1007/s00024-016-1407-8

Zaytsev, O., Rabinovich, A.B., & Thomson, R.E. (2017), The 2011 Tohoku tsunami on the coast of Mexico: A case study, *Pure and Applied Geophysics*, *174*(8), 2961-2986. https://doi.org/10.1007/s00024-017-1593-z

Zaytsev, O., Rabinovich, A.B., & Thomson, R.E. (2021), The impact of the Chiapas tsunami of 8 September 2017 on the coast of Mexico. Part 1: Observations, statistics, and energy partitioning, *Pure and Applied Geophysics*, *178*(11), 4291-4323. https://doi.org/10.1007/s00024-021-02893-x

Alexander B. Rabinovich